# Non-specific protection from respiratory tract infections in cattle generated by intranasal administration of an innate immune stimulant

**William Wheat**[1], **Lyndah Chow**[1], **Vanessa Rozo**[1], **Julia Herman**[1], **Kelly Still Brooks**[1], **Aimee Colbath**[1], **Randy Hunter**[2], **Steven Dow**[1] *

**1** Department of Clinical Sciences, From the Center for Immune and Regenerative Medicine, College of Veterinary Medicine and Biomedical Sciences, Colorado State University, Ft. Collins, Colorado, United States of America, **2** Hunter Cattle Company, Wheatland, Wyoming, United States of America

* sdow@colostate.edu

**Data Availability Statement:** All relevant data are within the paper and its supporting Information files.

**Funding:** These studies were supported by grant 1011 from the State of Colorado Office of

## Abstract

Alternatives to antibiotics for prevention of respiratory tract infections in cattle are urgently needed given the increasing public and regulatory pressure to reduce overall antibiotic usage. Activation of local innate immune defenses in the upper respiratory tract is one strategy to induce non-specific protection against infection with the diverse array of viral and bacterial pathogens associated with bovine respiratory disease complex (BRDC), while avoiding the use of antibiotics. Our prior studies in rodent models demonstrated that intranasal administration of liposome-TLR complexes (LTC) as a non-specific immune stimulant generated high levels of protection against lethal bacterial and viral pathogens. Therefore, we conducted studies to assess LTC induction of local immune responses and protective immunity to BRDC in cattle. In vitro, LTC were shown to activate peripheral blood mononuclear cells in cattle, which was associated with secretion of INFγ and IL-6. Macrophage activation with LTC triggered intracellular killing of *Mannheimia hemolytica* and several other bacterial pathogens. In studies in cattle, intranasal administration of LTC demonstrated dose-dependent activation of local innate immune responses in the nasopharynx, including recruitment of monocytes and prolonged upregulation (at least 2 weeks) of innate immune cytokine gene expression by nasopharyngeal mucosal cells. In a BRDC challenge study, intranasal administration of LTC prior to pathogen exposure resulted in significant reduction in both clinical signs of infection and disease-associated euthanasia rates. These findings indicate that intranasal administration of a non-specific innate immune stimulant can be an effective method of rapidly generating generalized protection from mixed viral and bacterial respiratory tract infections in cattle.

## Introduction

Bovine respiratory disease complex (BRDC) continues to be a major cause of economic losses in the cattle industry, despite improvements in transportation and feedlot management [1] [2] [3] [4] [5]. Moreover, recent government mandates and public pressure are increasingly

Economic Development and International Translation (OEDIT) and by grant COLV 2018-06 from Colorado State University Research Council (CRC) to SD. The OEDIT funding is provided to support the state of Colorado bioscience industry and funds were used to supply materials and salary support, including sample collection, research supplies, data analysis and publication costs. OEDIT and CRC funds supplied salary support for WW but the study sponsors did not have any additional role in the study design, data collection analysis, decision to publish, or preparation of the manuscript and only provided financial support in the form of the author's salary and/or research materials. Animal BRDC challenge studies were performed by Elanco, Inc. Elanco Inc covered the costs associated with animal procurement, animal boarding, and costs associated with the completing the animal challenge studies and assisted with data collection and analysis. The Hunter Cattle Company provided healthy cattle and blood samples for testing the immunological effects of LTC administration. Neither of these two commercial entities (i.e., Elanco Inc or Hunter Cattle Co.) had a role in supplying salary support, decisions to publish, or preparation of the manuscript. Elanco Inc did however design the animal challenge studies, in collaboration with other study investigators. The specific roles of each study author are articulated in the "author contributions" section.

**Competing interests:** SD, AC and RH hold stock options and corporate positions at LaPorte Ag Therapeutics, Inc, a Colorado State University-based startup company developing the LTC immunotherapy platform technology. WW, LC and SD are patent holders for the LTC technology. The issued US patent covering this technology is US 10,512, 687, issued Dec 24, 2019. Any commercial affiliation of the above-mentioned authors with LaPorte Ag Therapeutics, Elanco Inc. or Hunter Cattle Co. did not play a role in the study design, data interpretation, or publication decisions. Accordingly, this does not alter our adherence to PLOS ONE policies and sharing data and materials.

restricting the use of antibiotics for prophylaxis or metaphylaxis of BRDC [6]. Given these accelerating changes in the industry, there is a growing awareness of the need for non-antibiotic alternatives to generate broad-spectrum, non-specific protection from bacterial and viral pathogens associated with BRDC in cattle. Among non-antibiotic alternatives evaluated previously with variable success for disease prevention include phage therapy, antimicrobial peptides, and probiotics [7] [8] [9] [10] [11] [12] [13]. Recently, a liposome-TLR-9 agonist immunotherapeutic (Zelnate[R]) originally developed in our laboratory has been approved for parenteral administration in cattle for prevention of BRDC [14] [15] [16] [17] [18]. However, the overall effectiveness of most of these diverse approaches for BRDC prevention has yet to be fully demonstrated in challenge or clinical studies, though Zelnate has demonstrated modest activity in BRDC trials [14] [19]. An immunotherapeutic consisting of a combination of TLR and NOD-like receptor agonists (Amplimune[R]) has demonstrated activity in the prevention of calf scours, but activity against BRDC has not been reported [20]. Thus, there is clearly a need for new, non-antimicrobial approaches for prevention of BRDC.

For rapid induction of non-specific protection from bacterial and viral respiratory tract pathogens, activation of host innate immune defenses locally at mucosal surfaces in the nasal cavity and oropharynx has certain inherent advantages. For example, activation of upper airway innate immune defenses can suppress pathogen replication and invasion at the initial site of entry and provide significant protection from mortality, as has been demonstrated previously by our group in multiple rodent bacterial and viral challenge models using a liposomal-TLR-9 agonist immunotherapy [21] [22] [23] [24] [25] [26]. We also recently demonstrated in a cat model of feline herpesvirus-1 (FHV-1) respiratory tract infection that intranasal administration of L:TC (which consists of liposomes complexed to dual TLR-3 and TLR-9 agonists) could activate mucosal innate immune responses in the upper airways and significantly suppress viral replication [27] [28]. Additionally, the LTC immunotherapeutic activated innate immune responses in the oropharynx of dogs following intranasal delivery and significantly reduced clinical signs associated with canine herpesvirus infection [29].

To assess the potential for the LTC immunotherapeutic to activate upper airway immune defenses in cattle, we measured activation of innate immune responses using in vitro and in vivo studies. Immune activation and induction of antimicrobial activity in vitro was assessed using peripheral blood mononuclear cell (PBMC) and macrophage cultures. In healthy cattle, intranasal administration of LTC at 3 different doses was evaluated for local induction of innate immune activation in the nasal cavity and nasopharynx. Finally, the ability of intranasally-delivered LTC to induce protective antiviral and antibacterial immunity was assessed in a BRDC model of cattle exposed concurrently to bovine herpesvirus-1 (BHV-1/IBR), bovine diarrhea virus (BVD), and *M. hemolytica*.

These studies revealed that LTC induced rapid cellular activation in cattle, characterized by cytokine production and upregulated expression of immune co-stimulatory molecules. Intranasal LTC delivery to cattle triggered activation of local nasopharyngeal immune responses, while pre-treatment with LTC induced a significant reduction in clinical signs and mortality in a realistic BRD exposure challenge model. We concluded therefore that cattle responded immunologically to LTCs administered in the upper airways, and that LTCs have potential for use as a non-antibiotic solution for prevention or early amelioration of infections associated with BRDC.

## Materials and methods

### Preparation of liposome-TLR complexes

Liposomes were prepared as described previously [27] [29]. Polyinosinic-polycytidylic acid (InVivoGen, San Diego, CA) and plasmid DNA (non-coding commercial plasmid PCR2.1,

Thermo Fisher Scientific, Waltham, MA) were added to pre-formed liposomes to form liposome-TLR complexes (LTC). The endotoxin content of the plasmid DNA was between 0.04 and 0.25 EU/ug. Carboxy-methylcellulose (Sigma-Aldrich, St. Louis, MO) was added to the pre-formed complexes to produce the final LTC material for study.

## Cell culture medium

Peripheral blood mononuclear cells (PBMC) obtained from healthy cattle were cultured in complete medium, which consisted of DMEM (Thermo Fisher Scientific, Watham, MA) containing 15% FBS (Avanti, Alabaster, AL) and essential and non-essential amino acids and penicillin and streptomycin (Gibco and Thermo Fisher Scientific, Pittsburgh, PA).

## In vitro assays of immune activation

Whole blood for PBMC cultures was obtained by jugular venipuncture from healthy cattle and collected into EDTA tubes. All studies involving blood collection from healthy animals were approved by the Institutional Animal Care and Use Committee at Colorado State University. For separation of PBMC, blood was diluted 1:2 with sterile PBS, then layered over a Ficoll (GE Healthcare, Uppsala, Sweden) gradient and centrifuged. Cells were collected and washed twice in PBS and then re-suspended in complete tissue culture medium. Cells were plated in 96-well flat bottom plates (CellTreat, Pepperell, MA) at a density of $1 \times 10^6$ cells/well in 200μl of medium. For assays involving LTC activation of PBMC, LTCs were added at 3 different dilutions (1 μl per well, 0.3 μl per well, and 0.1 μl per well) in triplicate wells of PBMC in 100 μl complete medium, with careful mixing, and the cells were then incubated for an additional 48h. Cells were immunostained for detection of intracellular IFNγ expression using a mouse anti-bovine IFNγ antibody (Bio-Rad, Hercules, CA) and the percentage of IFNγ+ cells was determined by flow cytometry. Additionally, conditioned medium was collected for measurement of IFNγ secretion by ELISA and cells were collected for flow cytometric analysis of activation markers (see below) and for analysis of cytokine gene expression by quantitative real-time PCR (qRT-PCR) (see below). These assays were repeated at least twice, using PBMC from different donor animals.

## Analysis of cytokine gene expression by qRT-PCR

Nasopharyngeal samples collected from cattle before and after intranasal treatment with LTC (see below for details) were analyzed for changes over time in expression of IFNγ, IL-8 and MCP-1 using qRT-PCR. Briefly, cDNA was prepared from swabs of the nasopharynx following isolation of RNA from recovered cells, which was subsequently reversed transcribed using a commercial kit (Qiagen, Germantown, MD). Cellular cDNA was amplified using a qPCR MX300p system instrument (Agilent, Santa Clara, CA). Primer sequences for bovine cytokines were obtained from BLAST searches and qRT-PCR was used to quantify transcript levels, as reported previously [27] [29].

## Generation of monocyte-derived macrophages (MDM)

Peripheral blood mononuclear cells were isolated as described above and added to triplicate wells of 24-well plates at a concentration of $2 \times 10^6$ cells/ml for 2h at 37°C to allow monocyte adherence. Non-adherent cells were then washed off gently using PBS, and the wells re-fed with complete medium containing 10 ng/ml recombinant human M-CSF (PeproTech, Rocky Hill, NJ). After 7 days in culture, macrophages were fully differentiated from monocytes and then used in assays to examine the impact of LTC treatment on activation and bacterial killing.

## Bacterial strains

A clinical isolate of *Staphylococcus aureus* (strain 50804) was obtained from milk from a cow with clinical mastitis and provided by the CSU Diagnostic Laboratory. Clinical BRD isolates of *Mannheimia hemolytica* (strain 44168) and *Pasteurella multocida* (strain F180022916) were kind gifts of Dr. Josh Daniels (CSU Diagnostic Laboratories and the Department of Microbiology, Immunology, and Pathology). Staphylococci and pathogenic BRD strains were propagated in brain heart infusion (BHI) medium (Becton Dickinson, Franklin Lakes, NJ) in short-term culture, and bacteria were utilized in log phase of growth at a CFU density determined by comparison to a standard optical density curve. All pathogenic strains of bacteria were cultured in BHI and utilized at log phase of growth.

## Assessment of macrophage phagocytosis

Macrophage phagocytosis was performed using two assays: bead phagocytosis and bacterial phagocytosis. For analysis of polystyrene bead phagocytosis (FluroSpheres®; Thermo Fisher Scientific, Waltham, MA), MDM were detached from culture dishes by adding ice cold PBS containing 3 mM EDTA and allowing the plates to remain on ice for 20 min, and washed then re-plated at 3 X $10^5$ cells in a 24-well plate and allowed to reattach, and then incubated at 37°C with beads in complete DMEM, at a density of 200 beads per cell as described previously [30]. MDM were allowed to phagocytose the beads for 2 h at 37°C followed by removal of the medium and washing with 500 ul of 2%BSA in PBS. The bead-treated MDM were then detached using trypsin and analyzed for bead uptake (as measured by geometric mean fluorescent intensity) by flow cytometry and compared to untreated MDM.

For analysis of phagocytosis of bacteria, MDM were removed and counted as described above and re-plated and infected at a MOI = 5 with *S. aureus* strain Xen36 (Perkin Elmer, Waltham, MA). MDM were treated for 1 h with the bacteria in antibiotic-free medium at 37°C. Subsequently, the macrophages were washed twice with FACS buffer (PBS with 2% FBS and 0.05% sodium azide). The cells were then washed and permeabilized with 1:1000 diluted Triton X 1000 (1 μl/ml) for 10 min at 37°C. Cells were subsequently blocked with 10% goat serum in PBS followed by washing 2X in PBS containing 0.05% (V/V) Tween-20. Permeabilized cells were stained with a primary rabbit-anti-*S aureus* antibody for 1 h followed by washing and stained with a secondary goat-anti-rabbit-Cy3 antibody. Uptake of the bacteria was determined by assessment of geometric mean fluorescence intensity (gMFI) of Cy3 positive events using via flow cytometry.

## Nitric oxide (NO) release assay

The Griess reagent assay system (Promega, Madison, WI) was used to measure nitric oxide (NO) release by MDM by assessing formation of nitrite ($NO_2^-$) using culture supernatants, according to manufacturer's recommendations.

## Macrophage bactericidal assays

Bacterial uptake and killing by MDM was done as described previously by Drevets *et al.* [31]. Briefly, macrophages were incubated with LTC (1 ul per ml medium) for 24h, then cells were washed 3 times with fresh antibiotic-free medium. The effective concentration of LTC was determined by prior in vitro titration. Macrophage bactericidal activity was assessed by addition of *S. aureus* to cultured macrophages at a MOI of 5 (bacteria to macrophage). Bacteria were incubated with macrophages for 30 min, at which time remaining bacteria in supernatants were removed by washing with PBS. Assessment of bactericidal activity against *M.*

*hemolytica* and *P. multocida* was done similarly. Following incubation with bacteria, macrophages were cultured an additional 1h, at which time bacteria in supernatants were removed by washing with PBS. The macrophages were then lysed adding ice cold sterile $dH_20$ and vigorous pipetting, and concentrations of viable intracellular bacteria determined by quantitative plating of serially diluted samples on quadrant plates containing LB or BHI agar (or blood agar for *M. hemolytica*). As a positive control for induction of bactericidal activity, macrophages were incubated for 24h with 10 ng/ml bovine IFNγ (R&D Systems, Minneapolis, MN) prior to bacterial inoculation. Experiments were repeated at least twice using MDM obtained from unrelated animals.

## Administration of LTC to healthy cattle and assessment of immune responses

Studies in healthy cattle were completed at a stocker operation in Wheatland, Wyoming. These studies were approved by the Veterinary Research and Consulting Services Institutional Animal Care and Use Committee. Upon arrival at the ranch, health and welfare assessment of the animals was determined using standard practices including visual determinations of nasal discharge, ocular discharge, diarrhea, bloated rumen and external lesions such as skin lesion (hairless patches, subcutaneous edema, broken tails and ocular (sclera, cornea and lens) discoloration. Cattle having any of these lesions were not included in the study. Stocker animals of similar age (4–6 months) upon arrival at the ranch and were separated into groups of n = 20 to 30 animals each and then randomly assigned to treatment groups of n = 5 animals per group for immunological studies. Animals were treated by intranasal administration (using a teat canula) of either sterile PBS (control group, Group 1) or 1 ml LTC per nostril (Group 2), 2 ml LTC per nostril (Group 3), or 3 ml LTC per nostril (Group 4). Test material was administered intranasally to animals confined briefly in a head chute, using a teat canula attached to a Henke Roux syringe. In subsequent studies (see Figs 9 and 10), LTC were administered intranasally via a mucosal atomization device (MAD® LMA Teleflex Medical, Research Triangle, NJ) rather than a teat canula, based on studies demonstrating superior immune activation following MAD delivery. Animals were monitored before and after treatment, for a period of 14 days, with samples collected immediately prior to treatment, and again at 8 hours, 24 hours, 72 hours, 7 days, and 14 days after a single LTC treatment. Body temperature was evaluated at each monitoring time point.

## Nasopharyngeal mucosal cell sampling

Nasopharyngeal swabs were collected by manually inserting a sterile mare double-lumen swab (McCullough swab, Mountain Veterinary Supply, Ft Collins, CO) into one nare to the level of the nasopharynx, at which point the catheter was rotated against the nasopharyngeal mucosa to collect cells, then withdrawn and immediately placed into complete tissue culture medium on ice in a 15 ml conical tube. Any samples which contained obvious blood (< 5% of total samples) were discarded. The procedure was repeated on the opposite nare, and the two catheters were placed into the same 15 ml conical tube. Once in the laboratory, the swabs were vortexed together to free attached cells, which were pooled. The pooled cells were then split and used for flow cytometric analysis and for analysis of cytokine gene expression.

## Flow cytometry to assess cellular immune responses in vitro

For in vitro analysis of cell activation by LTC, PBMC were incubated with various concentrations of LTC for 48h, then were resuspended in flow cytometry buffer (PBS, 5% FBS, and 0.01% sodium azide) and immunostained using directly conjugated antibodies for flow

cytometric analysis. The following antibodies were used to enumerate immune responses by T cells, B cells and monocyte/macrophages: T cells: mouse anti bovine CD4-RPE (Bio-Rad, Hurcules, CA)/ mouse anti-bovine CD8-FITC (Bio-Rad)/ mouse anti-bovine CD5-APC (Bio-Rad); B cells: mouse anti-bovine CD21; monocyte/macrophages: cross-reactive mouse-anti-human CD14-PacBlue (clone Tük4; (Bio-Rad)/mouse-anti-bovine MHCII-FITC (BioRad). After immunostaining, cells were resuspended in flow cytometry buffer, and analyzed using a Beckman-Coulter Gallios multi-color flow cytometer. Flow cytometry data were analyzed using FlowJo® software (FlowJo, Ashland, OR).

## Impact of mucosal immunotherapy on the local microbiome

Studies were done next to assess the impact of intranasal immunotherapy with LTC on the endogenous nasal microbiome. For this study, nasal swab samples were collected from the same study groups (n = 10 animals per group, PBS or LTC treated) used to assess humoral immune responses (below). Nasal swabs (single swab per animal per collection time point) were collected from each animal using sterile cotton tipped applicators (Puritan Medical Products, Guilford, ME). Swabs were collected immediately prior to treatment and again at 7 days and 14 days after treatment. Swabs were stored in phosphate buffered saline at -20˚C until processing for extraction. Microbial DNA extraction was performed using a MoBio Powersoil DNA Isolation kit (Qiagen, Valencia, CA) according to manufactures' instructions. All samples were extracted using the same kit lot. Extracted DNA was submitted to Novogene Corporation (Chula Vista, CA) for 16S rRNA sequencing. Negative controls were verified on Nanodrop 1000 to have <2 ng/uL of total DNA. According to Novogene's report of analysis, DNA concentration and purity was monitored on 1% agarose gels.

## DNA library preparation and 16S sequencing

16S rRNA genes of V4 region were amplified using V4: 515F-806R in accordance with the Earth Microbiome project [32]. All PCR reactions were carried out with Phusion High-Fidelity PCR Master Mix (New England Biolabs, MA). PCR products were purified with Qiagen Gel Extraction Kit. Sequencing libraries were generated using TruSeq DNA PCR-Free Sample Preparation Kit (Illumina, San Diego, CA) following manufacturer's recommendations and index codes were added. The library quality was assessed on the Qubit@ 2.0 Fluorometer and Agilent Bioanalyzer 2100 system. The library was sequenced on an Illumina HiSeq 2500 platform and 250 bp paired-end reads were generated. Sequences were demultiplexed and forward and reverse pair-end reads were uploaded by Novogene.

## 16S Data analysis

Paired-end reads were merged using paired end demux [33]. Sequence quality control, adapter trimming and feature table construction were performed according to the QIIME2 version 2018.2 Demux Summarize DADA2 [33]. Based on quality score of 30. Operational Taxonomic Units (OTUs) were conducted at 97% sequence similarity using QIIME for taxonomically classified [34]. For genus level, the Green-Genes 16S rRNA gene database was used at 0.8 confidence threshold for taxonomic assignment. OTUs abundance information was normalized using a standard of sequence number corresponding to the sample with the least sequences (110,000). Phylogenetic tree was constructed using Qiime2 phylogeny fast tree [35]. Alpha diversity and beta diversity was calculated using Qiime2 diversity core metrics [36], while differential abundance testing was performed using Analysis of Composition of Microbiomes (ANCOM) [37]. Significance in relative abundance on Phylum, Class, Order, Family and

Genus levels was calculated using 2-way ANOVA with Bonferroni post-test. Graphical results were plotted using Graph Pad Prism8 (GraphPad Software, La Jolla California USA).

## Impact of LTC treatment on humoral immune responses to pathogenic bacteria

Studies were done to assess the impact of non-specific induction of mucosal immune activation in cattle on humoral immune responses against common bovine pathogens (*Mannheimia spp.*, *Pasturella spp.*, *Histophilus spp.*). There were two study groups of n = 10 animals each (weanling beef cattle), one of which was treated once with PBS (2 ml per nostril) and the other treated once with LTC, 2 ml per nostril. Blood was collected from each animal immediately prior to treatment, and again 30 days later, and serum prepared and frozen at -20 C˚ prior to analysis. For analysis of anti-bacterial antibody responses, bacteria (1.0 X $10^6$ bacteria per well) were added to individual wells of 96-well plates containing serum pre-diluted at a 1:100 dilution in FACS buffer. Bacteria were incubated for 30 minutes at room temperature, washed twice in FACS buffer, incubated with pre-diluted anti-bovine IgG (FITC conjugated, BioRad, St. Louis, MO) for 30 minutes, washed and resuspended in FACS buffer prior to analysis. Samples were analyzed using a Becton Dickenson (Franklin Lakes, NJ) Gallios flow cytometer, and a minimum of 50,000 events were collected for analysis. Negative controls included bacteria only and bacteria incubated with secondary antibody only, which was used to set analysis gates. Single color histograms were generated and used to determine percentage IgG$^+$ cells and the geometric mean fluorescence intensity of these cells, using FlowJo software.

## BRDC challenge study in cattle

Weaned 3- to 5-month-old, Holstein steers were procured from commercial vendors for the study and underwent a 5-day acclimatization period before being enrolled in the study. To ensure the animals experience a minimum amount of suffering and distress and are handled by staff with special training in research animal care, the study was conducted according to the Ag Guide 2010, and the protocol was approved by an Institutional Animal Care and Use Committee at Elanco, Inc using protocol number EIAC-0986. There were 3 groups of calves in the study: 1) Seeder animals (used to infect calves in Treatment Groups 1 and 2); 2) Treatment Group 1 animals (intranasal PBS administration), and 2) Treatment Group 2 animals (intranasal LTC administration). Only healthy animals were used in the study. Seeder calves and treatment calves were excluded from study if they were found to be BVDV positive, if they had received an antibiotic within 21 days of study day initiation, or if they had evidence of lung pathology via transthoracic ultrasound. All seeder calves were serologically negative for both IBR and BVDV 1b. Contact animals had no history of being administered any antiviral or antibacterial BRDC vaccines, including vaccines for IBR, BVDV, BRSV, PI-3, or common BRD bacterial pathogens. To ensure both the health and assess the pathogen burden status of contact animals prior to comingling with seeders, Elanco Inc. performed standard pathogen analysis as part of their IACUC protocol (EIAC-0986). This protocol included collection of nasal swabs that were plated on selective medium plates to enumerate bacterial pathogens. Swab samples were also analyzed for IBV and BVDV1b template levels by qRT-PCR. No significant growth of *M. haemolytica* was detected on plates from any animal prior to co-mingling with seeders. Additionally, all pre-exposed animals were negative for either IBR or BVDV1b by qRT-PCR.

Treatment animals were randomized to one of two treatment groups (n = 24 animals per group) and then co-mingled with seeder animals. Four groups of seeder animals were inoculated with pathogens intranasally prior to comingling with treatment animals. Seeder animals

were intranasally challenged with BHV-1 alone, BVD alone or BHV-1 plus *M. haemolytica*, or BVD plus *M. haemolytica*. Seeder animals were comingled with treatment animals throughout the 24-day study period. Treatment group animals were treated 24 h before they were first comingled with seeder animals. Treatment Group 1 calves were treated by intranasal injection of PBS to both nostrils (2 ml per nostril). Treatment group 2 calves were treated by intranasal administration of LTC (2 ml per nostril).

Animals in the two treatment groups were monitored daily for clinical signs by study personnel who were blinded to study groups. Serum samples were collected for cytokine assay (IL-6, TNFα, using commercially available ELISA kits from R & D Systems, Minneapolis, MN) on study days 0, 2, 4, 8, 12, 16, and 24. For study animals removed from the study after reaching pre-determined euthanasia endpoints defined as displaying open-mouth breathing, moderate to severe repeated coughing, no interest in approaching bunk and general moribund appearance an behavior. Euthanasia was performed within 12 hr of reaching any of these endpoints. None of the animals died before meeting euthanasia criteria. All remaining animals (48) were humanely euthanized on day 24 of the study and necropsies were performed and lung pathology scoring was performed to quantify grossly visible lung lesions known to be associated with *M. haemolytica* infection, including tissue consolidation, congestion and a fibrinous pleuropneumonia. Lesion scoring was performed by a clinical veterinarian blinded to treatment groups. In addition, at the time of necropsy, bronchial swabs were collected for cytokine evaluation by ELISA (R & D Systems, Minneapolis, MN).

## Results

### In vitro cellular activation by LTC

Initial studies were done to assess the ability of LTC to activate innate immune responses *in vitro*, using PBMC prepared from healthy cattle. Cells were incubated with the indicated amounts of LTC for 48h, at which time supernatants were collected for IFNγ and IL-6 assay by bovine specific ELISA. Cells from the same assay were collected for analysis by flow cytometry, as described in Methods. Addition of LTC to PBMC cultures triggered cellular innate immune activation in a dose-dependent fashion, as evidenced by increasing production of IFNγ with increasing doses of LTC (Fig 1A). Additionally, treatment of PBMC cultures with LTC increased expression of MHCII on CD14[+] monocytes, as assessed by flow cytometric analysis (Fig 1B). These immune responses are similar to those observed in prior studies following stimulation of mouse, canine, and feline cells with LTC [22] [24] [27] [29].

### LTC treatment of macrophages stimulates upregulated MHCII expression and secretion of TNFα and IL-6

Monocyte-derived macrophages (MDM) were evaluated for induction of immune activation via upregulation of MHCII following incubation with LTC. Following treatment, culture supernatants were also collected for analysis of TNFα and IL-6 release by ELISA. After supernatants were collected, treated cells were detached and immunostained for flow cytometric analysis. As illustrated in Fig 2A, LTC treatment of MDM stimulated significant upregulation of MHCII at LTC concentrations of 1.0 μl/ml and 5 μl/ml, and this response was observed using MDM cultures generated from 3 individual animals. Higher concentrations of LTC ($\geq$ 10 μl/ml) led to macrophage cytotoxicity. MHCII upregulation induced by LTC was comparable to that elicited by treatment with 20 ng/ml bovine IFNγ (Fig 2A). Production of TNFα and IL-6 by MDM was significantly increased when cells were treated with LTC, in a dose-dependent fashion (Fig 2B and 2C).

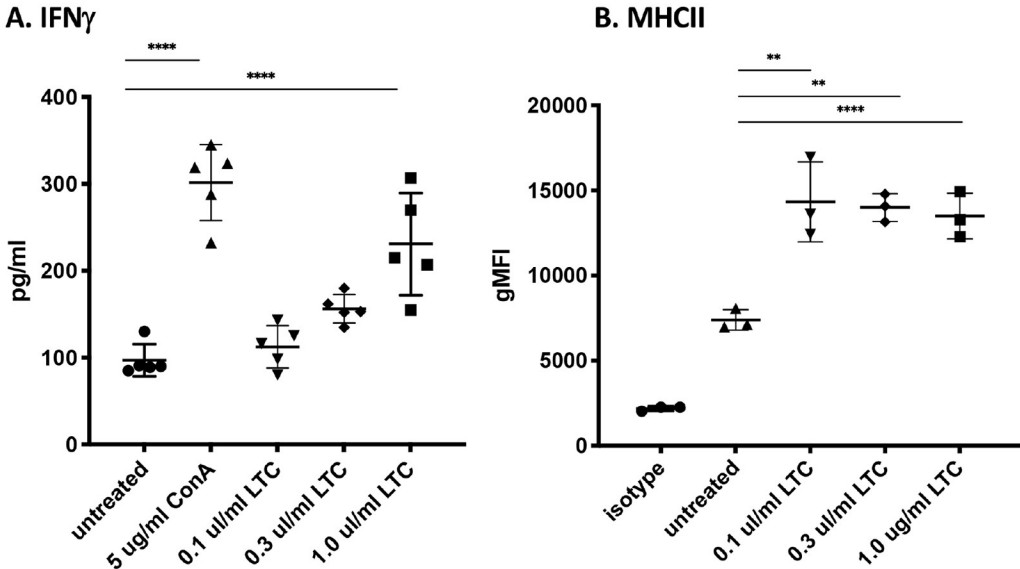

**Fig 1. Treatment of PBMC with LTC upregulates secretion of IFNγ and MHCII expression in a dose-dependent manner.** Peripheral blood mononuclear cells from 5 different healthy animals were cultured in vitro and treated with increasing concentrations of LTC for 48 h, as noted in Methods. Supernatants were collected and analyzed for IFNγ secretion by ELISA (**A**). Untreated or LTC-treated PBMC were collected and stained for T cells (CD3[+]) or monocytes (CD14[+]) and CD14[+] cells were analyzed for expression of MHCII (**B**). Similar results for MHCII expression were obtained for three separate experiments. Untreated or Concanavalin A (ConA) treated PBMC were used as negative and positive controls, respectively. Significant differences between either untreated animals or those either treated with ConA or LTC were determined using an ordinary one-way ANOVA with p values of ****, <0.0001 and **, <0.01.

## LTC treatment triggers macrophage nitric oxide (NO) release, phagocytosis and activation of bactericidal activity

Macrophage activation is generally associated with enhanced antimicrobial activity. To determine whether TLC activation also triggers these responses in macrophages from cattle, we assessed the response of MDM to LTC by evaluating NO release, phagocytosis, and bactericidal

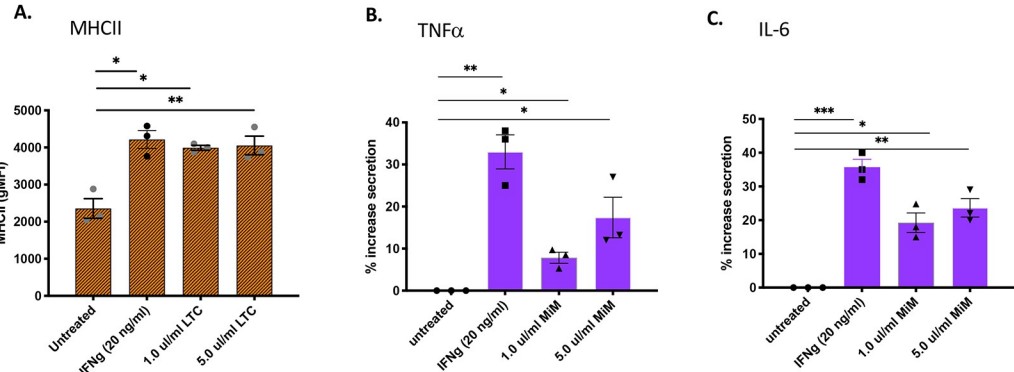

**Fig 2. Treatment of monocyte derived macrophages with LTC stimulates upregulation of MHCII expression and secretion of TNFα and IL-6.** Monocytes were prepared from PBMC from 3 different animals and differentiated for 7 days in vitro in 20 ng/ml human M-CSF. Monocyte-derived macrophages (MDM) were then treated for 36 h with LPS, INFγ, LTC, or LTC plus IFNγ at the indicated concentrations. MDM were analyzed for MHCII expression by flow cytometry and released cytokines were measured via ELISA. To normalize the absolute differences between animals, levels of secretion of TNFα and IL-6 were normalized to basal, unstimulated concentrations. Statistical analyses were performed using an ordinary one-way ANOVA with *, P<0.05, **, P<0.01, ***, P<0.005, ****, P<0.001 and 'ns', not significant.

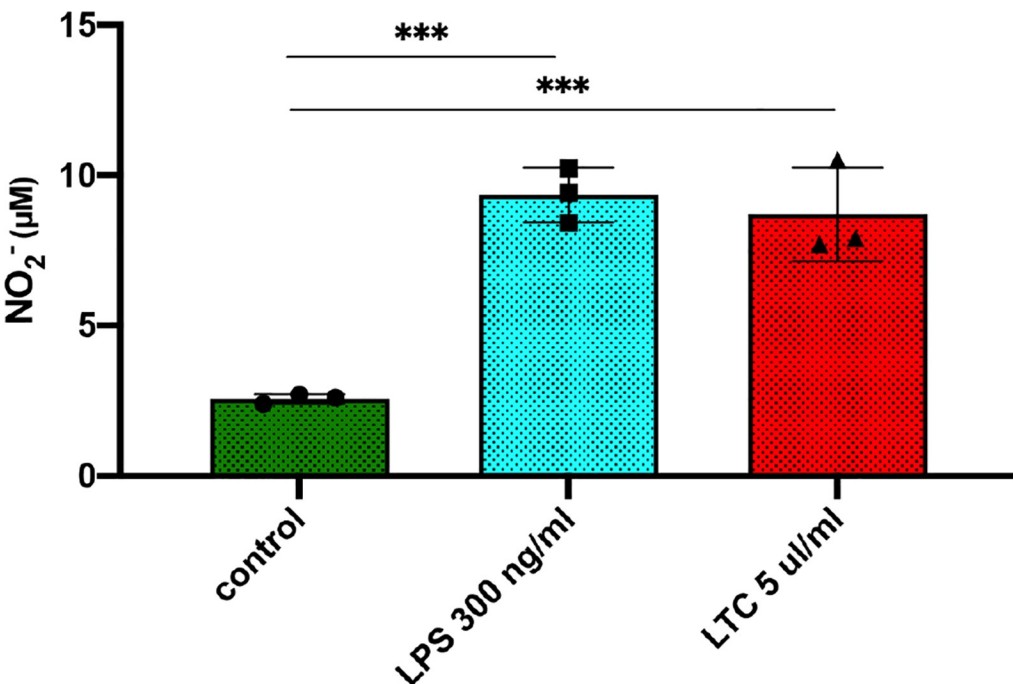

**Fig 3. LTC treatment stimulates nitric oxide release by macrophages.** Monocyte derived macrophages were treated for 36 h with either LPS or LTC at the indicated concentrations. Culture supernatants were assays for NO indirectly by measuring levels of nitrite ion ($NO_2^-$) release into MDM culture supernatants using the Griess reagent for detection of nitrate release. Significant differences were determined using an ordinary one-way ANOVA with; ***, P<0.005.

activity. In macrophages treated with LTC, there was a significant increase in release of NO (Fig 3). This response is important because production of reactive nitrogen intermediates is a key mechanism of intracellular bacterial killing. The impact of LTC treatment on macrophage phagocytosis was determined by measuring uptake of fluorescent beads and uptake of labeled *S. aureus*. LTC activated MDM exhibited significantly increased phagocytosis of both polystyrene FluorSpheres® and labeled *S. aureus* (S1 Fig). Thus, LTC treatment was found to activate two key functional antimicrobial properties of macrophages, namely phagocytosis and NO production.

Common bacterial pathogens in cattle include *S. aureus* (mastitis), and *M. haemolytica* and *P. multocida* (BRDC) [38] [39]. To assess the ability of LTC to activate macrophage bactericidal activity against these important cattle bacterial pathogens, MDM were pre-treated with LTC and induction of bactericidal was assessed. These studies revealed that LTC treatment of MDM generated significantly increased killing of all three pathogens. For example, LTC treatment increased killing of *S. aureus* by more than 100%, while killing of *M. haemolytica* and *P. multocida* was enhanced by over 100% and 10,000% respectively (Fig 4) compared to untreated macrophages. Pretreatment of MDM with the nitric oxide synthase inhibitor nitrosoguanidine (AG) prior to bacterial infection significantly reduced the amount of LTC-induced bactericidal activity, suggesting that NO production is an important mechanism for LTC-induced bactericidal activity (Fig 4).

## Mucosal immune activation following intranasal delivery of LTC

Healthy cattle (n = 5 per group) were treated with a single intranasal administration of LTC at 3 different doses (1 ml per nostril, 2 ml per nostril, and 3 ml per nostril), and their local upper

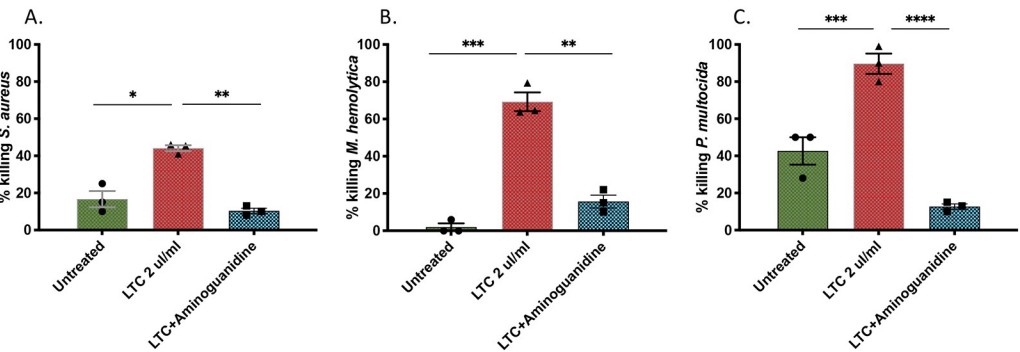

**Fig 4. Macrophage treatment with LTC triggers bactericidal activity.** Monocyte-derived macrophages from 3 animals were infected at a MOI = 5 for 3 h with *S. aureus* (**A**) *M. hemolytica* (**B**), or *P. multocida* (**C**), after which bactericidal activity was determined as noted in Methods. Each symbol represents one animal. Bactericidal activity was compared between untreated, LTC treated and aminoguanidine (AG) (10nM) pretreatment of LTC-treated macrophages. Comparisons between untreated, treated and AG pretreated cells for each bacterial pathogen were performed using an ordinary one-way ANOVA with *, P<0.05, ***, P<0.005.

respiratory tract immune responses compared to those of a control group of animals treated by intranasal administration of 1 ml PBS per nostril. Mucosal cell samples were harvested by swabbing the nasal cavity and nasopharynx as described in Methods and recovered cells were enumerated and analyzed by flow cytometry (Fig 5). We observed significant increases in overall upper respiratory tract cellularity in animals treated with 2 ml and 3 ml LTC per nostril at 24 h post treatment, compared to PBS treated control animals and compared to baseline pretreatment values (Fig 5A). Furthermore, significant increases in the percentages of CD14$^+$ monocytes were noted in animals treated at all 3 LTC doses, and the percentage of monocytes remained elevated at 72 h post treatment (Fig 5B). In addition, there was a concomitant increase in levels of expression of MHCII by monocytes from treated animals, compared to baseline values. Levels of MHCII expression peaked at 24 h post-treatment and then returned to baseline levels at 72 h post treatment (Fig 5C). Additionally, cells from nasopharyngeal swabs displayed significant increases in transcripts encoding IFNγ (Fig 6A), IL-8 (Fig 6B) and

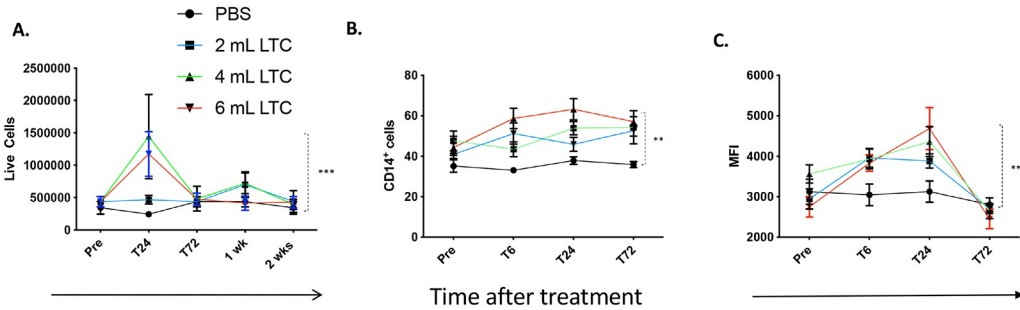

**Fig 5. Intranasal administration of LTC elicits cellular influx into the nasopharynx in a dose-dependent fashion in healthy cattle.** Four groups of healthy cattle (n = 5 per group) were treated intranasally with PBS (no treatment) or with 3 different dosages of LTC (1 ml per nostril (blue line), 2 ml per nostril (green line), or 3 ml per nostril (red line)). Nasopharyngeal swab samples were obtained at various intervals after treatment, and collected cells were analyzed by flow cytometry. Pharyngeal cells were also analyzed for increases in overall cellularity (**A**) and numbers of CD14$^+$ monocytes (**B**) and expression of MHCII (**C**). Comparisons of differences in percentages of the number of live cells, CD14$^+$ cells and mean fluorescence intensity (MFI) of MHCII between groups were evaluated as a function of time after treatment and dosage of LTC. Analysis of variance of these parameters was evaluated using a two-way ANOVA with, **P <0.010; and ***P<0.005.

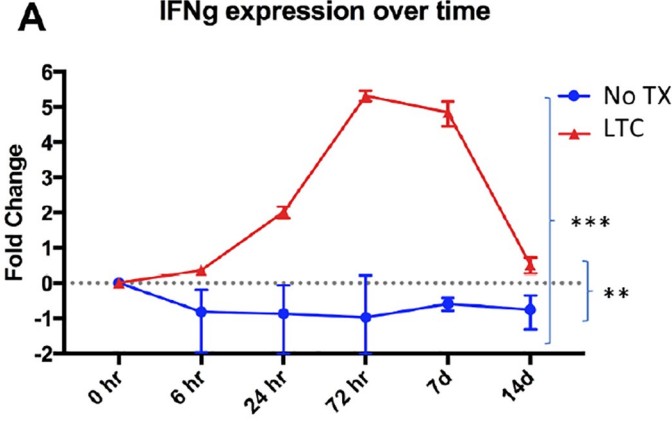

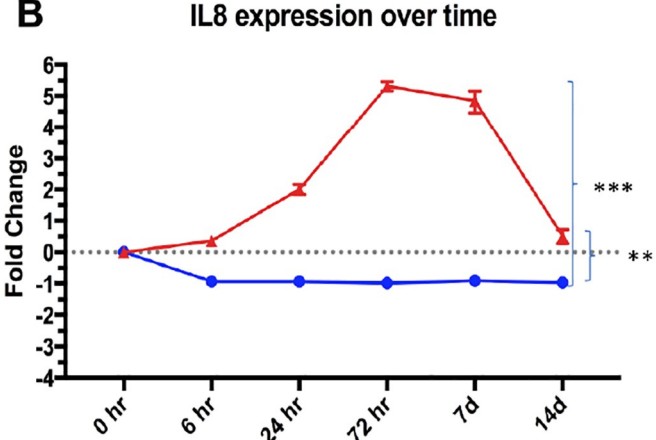

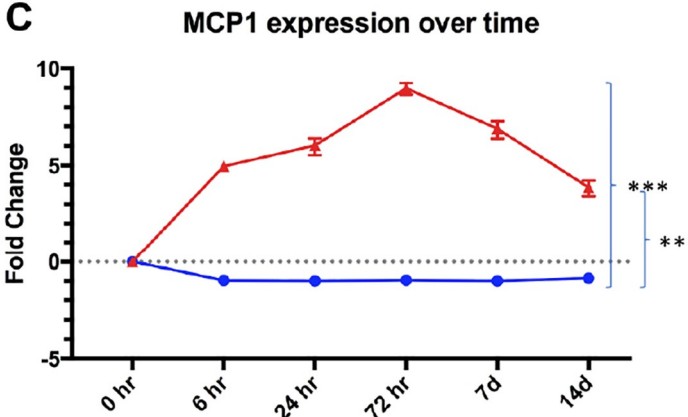

**Fig 6. Pharyngeal cells from LTC treated animals show increased expression of IFNγ, IL-8 and MCP-1 gene transcripts.** RNA was purified from nasopharyngeal swab samples obtained from cattle (n = 5 per group) treated with PBS or LTC (2 ml per nostril) at the indicated times post-treatment. cDNA was reverse transcribed and amplified by qRT-PCR. Transcript numbers were compared after 6h, 24h, 72h, 7days, and 2 weeks of treatment and plotted accordingly. Statistical differences in PBS-treated versus LTC-treated groups were evaluated using a two-way ANOVA with **P <0.010; and ***P<0.005.

MCP-1 (Fig 6C) as determined by quantitative reverse transcriptase PCR analyses of mRNA. As illustrated in Fig 6, there was a significant increase in cytokine gene transcription in cattle treated with 2 ml LTC per nostril, compared to PBS-treated control animals.

## Intranasal LTC treatment increased systemic humoral immune responses against pathogenic bacteria

To determine whether non-specific mucosal immune activation with LTC might also alter systemic humoral immune responses to bacterial pathogens known to be harbored in the nasal and pharyngeal mucosa in cattle, we evaluated humoral immune responses in cattle prior to and 30 days following a single intranasal administration of LTC (2 ml per nostril). Two study groups (n = 6 animals per group) of beef cattle were treated intranasally once with either PBS or LTC. Blood samples were collected prior to treatment and again 30 days after treatment and serum samples were analyzed for antibody responses against potential BRDC pathogens, including *M. haemolytica*, *P. multocida*, *and H. somni*. The serum IgG response to bacteria was assessed using a flow cytometric assay, as described in Methods. We found that the degree of IgG binding to all 3 species of pathogenic bacteria in 30d post-treatment serum samples from LTC treated animals was significantly greater than in pre-treatment samples, and also significantly greater than in control, PBS treated animals (Fig 7). These findings suggest that local administration of LTC complexes to the upper airways in cattle may trigger local activation of B cells specific for pathogenic bacteria, leading to greater serum IgG concentrations. Interestingly, the increased IgG response appeared to be specific for pathogenic bacteria, as we did not observe a greater IgG response against commensal, non-pathogenic bacteria.

## Nasal microbiome not significantly perturbed following LTC treatment

Activation of local innate immune responses has the potential to alter the local microbiome, given the strong reciprocal interaction known to occur between the immune system and the resident microbiome. To determine whether LTC administration would alter the nasal microbiome, we sampled nasal swabs from cattle (n = 10 animals per group) obtained prior to treatment and again 7 days following treatment with 2 ml LTC per nostril or 2 ml PBS per nostril. The composition of nasal microbiome was determined by 16S rRNA sequencing, as described in Methods. No significant differences between pre- and post-treatment groups were observed based on Simpson's evenness score (Fig 8A) or other diversity indices including Faith alpha

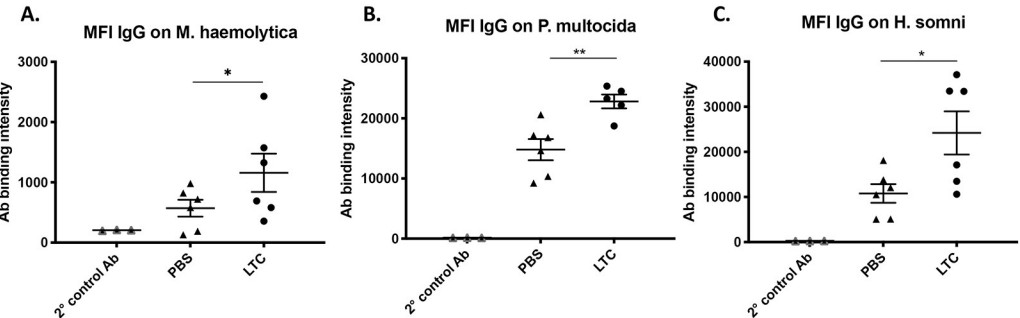

**Fig 7. Intranasal LTC treatment generates increased serum IgG binding to BRDC pathogenic bacteria.** Serum obtained from PBS treated or LTC treated (2 ml per nostril) cattle (n = 6 per groups) was incubated with BRDC bacteria in vitro, and IgG binding determined by flow cytometry, as described in Methods. Serum IgG binding to *M. hemolytica* (**A**), *P. multocida* (**B**) and *H. somni* (**C**) is depicted, comparing untreated versus LTC treated animals. Antibody binding intensity was displayed as geometric mean fluorescence intensity of IgG positive bacteria. Significant differences were detected using an unpaired t test with, *, P<0.05 and **, P<0.01.

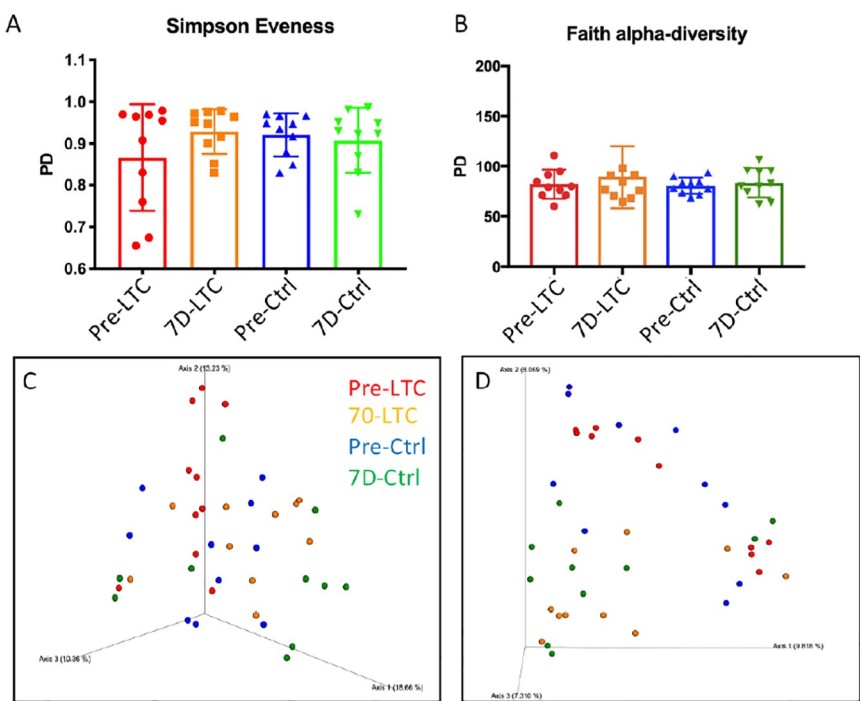

**Fig 8. Impact of intranasal immunotherapy on the nasal microbiome.** Bacteria were recovered from nasal swabs obtained from cattle (n = 10 per group) treated by intranasal administration of PBS or LTC (2 ml per nostril). Swabs were obtained pre-treatment and again at 7 days after treatment, bacteria were isolated and DNA extracted, and subjected to 16S sequencing at a commercial laboratory, as described in Methods. Analysis of sequence information revealed the following: (A) Diversity analysis of Simpson's evenness measure depicted on x-axis, y-axis represents phylogenic diversity (PD). Red bars represent pretreatment diversity in the LTC-treated group, orange represents 7 days after treatment in the LTC-treated group. Blue and green bars represent pre and 7 day valued in the control (PBS-treated) group. (1B) Faith alpha diversity. (1C) Principal components analysis of Bray-Curtis distance beta diversity (color legend in top right corner). Proportion of variance explained by each principal coordinate axis is denoted in the corresponding axis label. (1D) principal component analysis of Jaccard beta diversity.

diversity (Fig 8B), [40] Chao1, Pielou's Evenness, or Shannon or Simpson's alpha diversity indices. Moreover, no clear clustering of diversity differences in the nasal microbiome between LTC-treated and PBS-treated animals was observed using Bray-Curtis distance beta diversity analysis (Fig 8C). There was between group separation observed in Jaccard distance (Fig 8D), though it is possible that this separation could be due to sample collection time point implied differences rather than by LTC treatment per se. No features that were differentially abundant across the 4 groups of samples were found by ANCOM [37]. In addition, the lack of alterations in the nasal microbiome observed in the present study in cattle are consistent with recently reported findings from our group regarding the lack of impact of LTC treatment on the oro-pharyngeal microbiome of dogs [29].

## Impact of LTC immunoprophylaxis on clinical signs and disease-related euthanasia (mortality) following BRDC challenge in cattle

Finally, we evaluated the ability of LTC immunoprophylaxis to protect cattle from a realistic 3-pathogen challenge designed to mimic BRDC. Study animals (n = 24 animals per group) were exposed by co-mingling study animals with seeder animals that had been pre-infected by direct intranasal inoculation with either IBR, BVD, or *M. hemolytica*, as described in Methods. One group of study animals (control) was treated with PBS (2 ml per nostril) 24h prior to

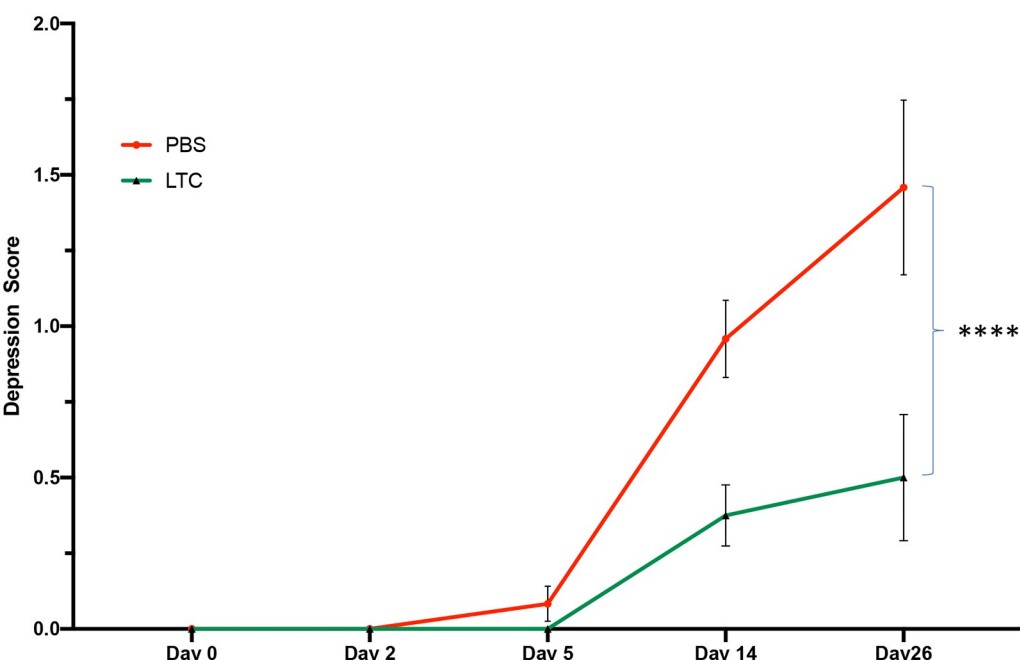

**Fig 9. Impact of LTC pre-treatment on clinical illness scores in cattle subjected to BRDC challenge.** Cattle (n = 24 per group) were treated with PBS (2 ml per nostril) or LTC (2 ml per nostril), and then 24 h later were co-mingled with BHV-1 infected, BVD infected, and *M. haemolytica*- infected seeder animals, as described in Methods. All exposed treated animals were monitored daily for clinical illness scores by blinded clinical observers, and overall scores tallied for each group of animals for the entire 24-day study period and mean and SEM values for illness scores were plotted. Data were analyzed for significance using a 2-way ANOVA with **, $P<0.01$.

comingling with infected seeder animals, while the second group (LTC) was pre-treated with intranasal administration of LTC (2 ml per nostril) 24h before comingling. Study animals were observed daily for development of clinical signs associated with BRDC, by scorers who were blinded to the treatment status of the animals. In addition, hematological and biochemical changes were monitored by every other day analysis of blood samples and cytokine concentrations (Il-6, TNFα) were monitored in blood samples as well.

Clinical monitoring revealed that animals treated prophylactically with LTC had significantly reduced clinical depression scores over the course of the 24-day study, when compared to animals treated similarly with PBS (Fig 9). For example, the clinical score was reduced from an average of 1.5 in PBS-treated animals to 0.5 in LTC treated animals, which represented a statistically significant overall reduction. Analysis of serum IL-6 and TNFα concentrations in PBS treated animals at day 0, day 8 and day 12 post-infection showed that concentrations of both cytokines increased marginally but not significantly overall by day 8 post challenge. However, in LTC treated animals, serum IL-6 concentrations were significantly reduced at day 8 and day 12 when compared to PBS treated animals. At the completion of the study, cytokine analysis of bronchial swabs collected at necropsy revealed significant reductions in airway IL-6 concentrations in PBS treated animals (81.34 pg/ml, [+/- 57]), compared to (42.77 pg/ml, [+/- 42]) in LTC treated animals (S2 Fig). These data indicated therefore that pre-treatment with LTC ameliorated the increase in pro-inflammatory cytokines associated with BRDC infection in cattle.

Additionally, culturing pathogenic respiratory bacteria procured from nasal swabs of the cattle before being exposed to seeder animals indicated that most of the cattle (62.5% of the PBS treatment group and 52.16% of the LTC-treated group) cultured no significant levels of

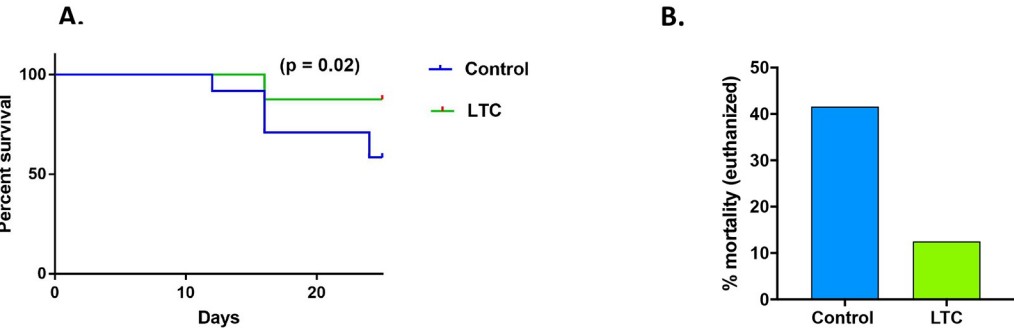

**Fig 10. LTC pre-treatment significantly reduces euthanasia (mortality) associated with experimental BRDC infection in cattle.** Cattle (n = 24 per group) were treated with PBS (2 ml per nostril) or LTC (2 ml per nostril), and then 24h later were co-mingled with BHV-1 infected, BVD infected, and *M. haemolytica*-infected seeder animals, as described in Methods. Animals in each treatment group were monitored daily for signs of clinical illness by observers and were euthanized when clinical scores reached a pre-determined score. Survival was evaluated and represented by Kaplan-Meier survival curve, followed by log-rank analysis to determine the level of statistical significance (p = 0.02) (**A**). In animals pre-treated with LTC, the total percentage of animals euthanized due to BRDC associated clinical illness was 12.5%, which was markedly lower than the 42% euthanized due to clinical illness for animals in the PBS treated group (**B**).

*Mannheimia spp*. *P. multocida* or *T. pyogenes*. At the end of the study (day 24) lung necropsy samples showed that 57% of the cattle from the PBS control group lung tissue cultured high CFU of *Mannheimia spp*. or P. *multocida* whereas only 14.2% of the LTC group scored in the high CFU category indicating a significant difference in pathogen burden. (S3 Fig).

Euthanasia due to clinical signs of BRDC infection (used here as a proxy for mortality) in the two groups of cattle were also assessed the course of the 24-day study. Observers scoring the cattle were blinded as to treatment groups to which the animals were assigned. Survival analysis by Kaplan-Meier curves revealed that survival was significantly increased (p = 0.02) in the LTC treated animals, compared to PBS treated animals (Fig 10A). In PBS-treated animals, 10 animals were euthanized due to clinical manifestations of severe BRDC-induced illness (Fig 10B). By contrast, in the LTC treated group, only 3 animals were euthanized due to BRDC-induced illness, thereby demonstrating a marked overall reduction in euthanasia (mortality) of 71%. These findings are important in that they provide critical evidence of the ability of effective immunoprophylaxis with a potent, non-specific mucosal immunotherapeutic to prevent or hinder the onset of clinically severe BRDC and associated death losses in cattle.

## Discussion

Bovine respiratory disease complex is a significant health problem in cattle throughout much of the developed world wherever intensive cattle husbandry is practiced [41] [42] [43] [44]. As such, BRDC is the principal source of economic loss for the North American beef industry and is also a significant source of losses in the dairy industry in early weaned calves [4] [45] [1] [46]. There are multiple factors associated with triggering BRDC in cattle, including complex interactions between the environment, host factors, and viral and bacterial pathogens [2] [3] [47]. Stressors to cattle can adversely affect both immune and nonimmune defense mechanisms. Thus, the pathogenesis of BRDC typically involves some combination of stressors interacting with altered host immunity and subclinical infection to trigger clinically apparent disease. For example, viral infection is known to compromise host defenses and stimulate deeper invasion of pulmonary tissues by pathogenic bacteria normally commensal in the nasopharynx—in particular members of the *Pasteurellaceae* family [48] [48] [49] [50].

To address the need for more effective, non-antimicrobial measures to prevent BRDC or to lessen disease severity, we have turned to local activation of innate immune defenses in the nasal passages and oropharynx. Previously, our laboratory has demonstrated the utility and effectiveness of intranasal delivery of potent innate immune activators such as liposome-TLR9 complexes for prevention of mortality due to lethal viral or bacterial infections [21] [22] [23] [24] [25] [26] [27] [28] [29]. Our work has demonstrated earlier that when TLR3 or TLR9 agonists are complexed to cationic liposomes, their potency for activating innate immune responses is significantly increased [16] [51]. For example, we found that liposomal delivery of TLR3 and TLR9 agonists was particularly effective for stimulating type I interferon responses and CD8 T cell cross-priming [51]. Prior work has also established that combinations of certain TLR agonists, especially TLR3 and TLR9 agonists, are capable of generating synergistic innate immune activation when co-delivered to the same antigen presenting cells [52] [53] [54] [55] [16] [17] [25] [56] [56] [57] [58] [59] [15]. Therefore, we reasoned that a combined TLR3/9 agonist liposomal immunotherapeutic, optimized for delivery to mucosal surfaces [29] [29] [29], might be effective as a non-specific mucosal immune stimulant in cattle, in particular for preventing or lessening clinical signs and mortality associated with BRDC.

To address this question, we evaluated the local immune stimulatory properties of LTC in cattle, incorporating both in vitro and in vivo studies. Initial in vitro studies revealed that LTC treatment of PBMC stimulated IFNγ production (Fig 1), indicative of activation of key innate immune pathways. Additionally, LTC treatment of macrophages triggered upregulated MHCII expression (most likely mediated by autocrine released TNFα) and increased TNFα and IL-6 cytokine release (Fig 2). The importance of these findings relates to the known key role of MHCII expression for more efficient antigen presentation by macrophages to T cells, while both TNFα and IL-6 are important components of innate immunity and regulation of anti-viral and anti-bacterial immune responses [60] [61] [62] [63]. Importantly, macrophage treatment with LTC also triggered significant intrinsic macrophage bactericidal activity, against 3 different important pathogens, including *S. aureus*, *M. hemolytica* and *P. multocida* (Fig 4). This macrophage bactericidal activity appeared to be related in part to LTC induction of reactive nitrogen intermediate production, as revealed by NO pathway inhibitors (Fig 4). The LTC-mediated increased release of TNFα in bovine MDM is significant since TNFα has been shown to induce high levels of bactericidal NO likely by inducing increased levels of transcription of iNOS mRNA in macrophages [64]. Additionally, TNFα is chemotactic for monocytes in vivo which is prerequisite for their inflammatory and innate immunity functionality [65].

Studies were done *in vivo* to determine whether topical delivery of LTC could trigger significant local innate immune activation. Prior studies of a related innate immune agonist (Zelnate®) had demonstrated immune activation following systemic (intramuscular) administration, but the mucosal effects of a similar immune stimulant have not been previously evaluated in cattle [14]. Importantly, we found that intranasal administration of even relatively low doses of LTC in healthy cattle triggered rapid immune activation at mucosal surfaces. For example, following intranasal LTC delivery, there was a rapid cellular influx into the nasopharyngeal region (Fig 5A). Specifically, LTC treatment appeared to preferentially trigger monocyte recruitment, with a significant dose-dependent recruitment of CD14$^+$ monocytes occurring in the nasopharynx within 6h of LTC administration and continuing for another 72h (Fig 5B). The recruited monocytes also exhibited upregulated MHCII expression, consistent with activation by either IFNγ or TNFα (Fig 5C). Activated monocytes are a critical component of innate immunity and provide a potential bridge between innate and adaptive immune responses, both by generating antimicrobial activity directly and by presenting viral and bacterial antigens effectively to stimulate T cell responses indirectly.

We also observed sustained upregulation of expression of innate immune response genes for at least 2 weeks in cells obtained from the nasopharynx of cattle following a single intranasal administration of LTC (Fig 6). For example, upregulated expression of IFNγ, IL-8 and MCP-1 transcripts were observed at 2 weeks following LTC treatment. These findings are important because they speak to the duration of immune activation elicited by LTC and suggest that immune stimulatory complexes designed to adhere more effectively to mucosal surfaces, as in the case of LTC used in these studies, can trigger sustained immune activation and hence potential protective immunity for several weeks.

Interestingly, we found that LTC administration to the nose did not significantly disrupt the nasal microbiome (Fig 8). This surprising finding was also observed in recently reported study in healthy dogs treated with LTC and suggests that the endogenous microbiome may be relatively resistant to transient immune perturbations induced by innate activation alone [29]. This finding is important, inasmuch as it is well-established that the microbiome is highly susceptible to significant disruption induced by antibiotic treatment [66] [67] [67]. Thus, an important advantage to the use of an immunotherapeutic approach to BRDC prevention is preservation of the normal flora, which serves as an important barrier to pathogen colonization [68].

Another intriguing finding was the stimulatory effect of local activation of innate immune responses in the upper respiratory tract on systemic humoral immune responses to several key bacterial pathogens often harbored sub-clinically in the upper respiratory tract of cattle [69]. Thus, in animals treated with a single intranasal LTC administration, we observed significant increases in IgG responses to *Mannheimia*, *Histophilus*, *and Pasteurella*, consistent with activation of specific T cell and B cell responses to these pathogens. Though the exact mechanism of this effect remains undetermined at present, it is possible that LTC in the nose and oropharynx may bind (via charge-charge interactions) to resident bacteria in those sites, thereby serving to create an in-situ vaccine against these pathogenic organisms. Notably, the response appeared to be pathogen specific, inasmuch as IgG binding to a non-pathogenic bacterium (eg, *Bibersteinia trehalosi*) was not increased following LTC treatment. Thus, local mucosal immune activation appeared to augment immune responses against potential mucosal bacterial pathogens.

The ultimate goal of a respiratory tract innate immune stimulant such as LTC would be efficient and rapid induction of non-specific protection against infection caused by a complex mixture of viral and bacterial pathogens, a key feature of the pathogenesis of BRDC in cattle. To determine whether LTC could in fact induce protective immunity against BRDC without use of antibiotics, a realistic BRDC challenge model was utilized. In this model, test animals exposed to "seeder" animals infected with IBR, BVD, or *M. hemolytica* rapidly developed clinical infection. Clinical illness scores in LTC treated animals were significantly reduced compared to scores in control, PBS-treated animals (Fig 9). Thus, the average clinical illness score declined from 1.5 to 0.5 in treated animals compared to control (PBS-treated) animals. Moreover, the euthanasia rate (proxy for mortality rate) was markedly reduced by LTC pre-treatment, from 42% in control animals (PBS treated) to 12% in LTC-treated animals, a 71% reduction in overall euthanasia rate (Fig 10). This degree of reduction in animal morbidity and mortality associated with BRDC in cattle would have a significant economic benefit to the producer, both by reducing animal losses outright, as well as by minimizing the reduction in weight gain associated with BRDC. Taken together, these data indicate that intranasal delivery of a potent innate immune stimulus, even without antibiotic treatment, can reduce morbidity and mortality associated with BRDC in cattle.

Prior studies in rodent models indicate that innate immune activation can protect hosts from a variety of infectious agents [70] [71] [71]. Such an effect induced by an immune stimulant would also be an important factor in increasing the overall resistance of younger and at-

risk animals to BRDC infection. Finally, the non-specific innate immune activation approach may also lessen emergence of antimicrobial resistance by bacteria in the respiratory and gastrointestinal tracts, and thereby reduce the overall pressure on producers and feedlot operators to use antibiotics for prophylaxis or metaphylaxis.

## Supporting information

**S1 Fig. Macrophage treatment with LTC stimulates increased phagocytosis.** Triplicate cultures of MDM were either untreated, treated with LPS or with increasing concentrations of LTC for 24 h, followed by addition of fluorescent 1 um beads, as described in Methods (**A**). Cells were incubated with beads for 2 h and harvested by trypsinization followed by flow cytometric analysis to quantitate bead positive cells. In separate studies, MDM were incubated with *S. aureus*, and intracellular uptake quantitated by immunostaining for *S. aureus*, followed by analysis by flow cytometry. Macrophages were pre-treated with vehicle, LPS or increasing concentrations of LTC followed by infection with *S. aureus*. Analysis of variance of gMFI parameters was evaluated using a two-way ANOVA with, *, $P<0.05$; **, $P <0.01$; ***, $P<0.005$ and ****, $P<0.0001$.
(TIF)

**S2 Fig. Cytokine concentrations in bronchial swabs from cattle treated with PBS or LTC prior to induction of BRDC.** Cattle were treated with PBS or LTC administered by the intranasal route (2 ml per nostril) prior to exposure to BRDC seeder animals. Bronchial swabs were obtained at necropsy in animals that were euthanized due to BRDC or study completion (24 days). Bronchial samples were analyzed for IL-6 concentrations by ELISA. Statistical comparisons of differences in cytokine release was analyzed by ordinary one-way ANOVA with multiple comparisons with, *, $P< 0.05$.
(TIF)

**S3 Fig. Cattle in contact with seeder cattle infected with *M. hemolytica*, IBR and BVDV1b treated with LTC have lower pathogen burden in upper and lower respiratory mucosa when compared to animals treated with PBS.** Nasal swabs were obtained from cattle and plated on Brain Heart Infusion or blood agar plates and CFU were analyzed and assessed for pathogenic strains of respiratory bacteria including *Mannheima spp.*, *P. multocida* and *Truperella pyogenes*. Extent of infection was assessed as either "no significant CFU"(**A**), "few/moderate CFU (**B**) or "high CFU" (**C**). For day 24, CFU were obtained from lung necropsies. Data were analyzed for significance using an unpaired t test with *, $P \leq 0.01$.
(TIF)

## Acknowledgments

The authors wish to thank students in the Center for Companion Animal Studies and Elanco, Inc. for their help with animal studies.

## Author Contributions

**Conceptualization:** William Wheat, Julia Herman, Kelly Still Brooks, Aimee Colbath, Randy Hunter, Steven Dow.

**Data curation:** William Wheat, Lyndah Chow, Vanessa Rozo, Julia Herman, Kelly Still Brooks, Aimee Colbath, Randy Hunter, Steven Dow.

**Formal analysis:** William Wheat, Lyndah Chow, Julia Herman, Steven Dow.

**Funding acquisition:** Steven Dow.

**Investigation:** William Wheat, Lyndah Chow, Vanessa Rozo, Julia Herman, Kelly Still Brooks, Aimee Colbath, Steven Dow.

**Methodology:** William Wheat, Lyndah Chow, Vanessa Rozo, Julia Herman, Kelly Still Brooks, Randy Hunter, Steven Dow.

**Project administration:** William Wheat.

**Resources:** Kelly Still Brooks, Randy Hunter.

**Supervision:** William Wheat, Julia Herman, Steven Dow.

**Validation:** William Wheat, Aimee Colbath.

**Visualization:** Randy Hunter.

**Writing – original draft:** William Wheat.

**Writing – review & editing:** Lyndah Chow, Aimee Colbath, Steven Dow.

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
