## [Decision Letter · Decision Letter 0]

28 May 2020

PONE-D-20-09236

Non-Specific Protection from Respiratory Tract Infections in Cattle Generated by Intranasal Administration of an Innate Immune Stimulant

PLOS ONE

Dear Dr. Wheat,

Thank you for submitting your manuscript to PLOS ONE. After careful consideration, we feel that it has merit but does not fully meet PLOS ONE’s publication criteria as it currently stands. Therefore, we invite you to submit a revised version of the manuscript that addresses the points raised during the review process.

1) The authors should provide some tests about the disease/health status of the animals before use in the assays;

2) Please, answer to all the questions raised by both the reviewers.

We look forward to receiving your revised manuscript.

Kind regards,

Paulo Lee Ho, Ph.D.

Academic Editor

PLOS ONE

Journal Requirements:

"SD was supported by a grant from the State of Colorado Office of Economic

Development and International Translation (OEDIT) and by a grant from Colorado State University Research Council (CRC)#COLV 2018-06. OEDIT is intended to

support the state of Colorado bioscience industry and funds were used herein to fund

materials and salary support for studies, including sample collection, research

supplies, data analysis and publication costs. Animal BRDC challenge studies were

supported by Elanco, Inc.

We note that one or more of the authors are employed by a commercial company: "Hunter Cattle Company".

We note that you received funding from a commercial source: "Elanco, Inc"

a) Please provide an amended Funding Statement declaring these commercial affiliations, as well as a statement regarding the Role of Funders in your study. If the funding organization did not play a role in the study design, data collection and analysis, decision to publish, or preparation of the manuscript and only provided financial support in the form of authors' salaries and/or research materials, please review your statements relating to the author contributions, and ensure you have specifically and accurately indicated the role(s) that these authors had in your study. You can update author roles in the Author Contributions section of the online submission form.

Reviewers' comments:

Reviewer's Responses to Questions

**Comments to the Author**

1. Is the manuscript technically sound, and do the data support the conclusions?

Reviewer #1: Yes

Reviewer #2: Yes

2. Has the statistical analysis been performed appropriately and rigorously? 

Reviewer #1: Yes

Reviewer #2: Yes

3. Have the authors made all data underlying the findings in their manuscript fully available?

Reviewer #1: Yes

Reviewer #2: Yes

4. Is the manuscript presented in an intelligible fashion and written in standard English?

Reviewer #1: Yes

Reviewer #2: Yes

5. Review Comments to the Author

Reviewer #1: In this study a series of in vitro and vivo assays were performed to demonstrate the potential of lipossomes complexed to TLR-3 and TLR-9agonists (LTC) to activate upper airway innate imune responses, protecting cattle of infections associated with Bovine Respiratory Disease Complex (BRDC).

The assays were performed in PBMC, PBMC derived macrophages and in naso pharingeal cells from healthy animals, where they demonstrated a dose response activation of the different cells by LTC incubation in culture. Several parameters were evaluated such as IFN-gamma, IL-8, IL-6, MCP1 secretion and/or gene expression of these mediators and increase of class II MHC expression by activated monocytes or macrophages

In PBMC derived macrophages from healthy donors they studied the action of LTC in phagocytosis of beads and of bacteria, as well as the Nitric oxid dependent bactericidal activity following LTC incubation. All these reactions were improved by LTC in a dose dependente manner.

All experiments were well designed and the results converge to the demonstration of the effectiveness of LTC as an alternative preventive or therapeutic agent to BRDC.

Our only concern is about the last in vivo experiment. There it is lacking the confirmation of the infectious status of the experimental groups that were in contact with the seeders carrying the pathogens. Clinical signs were scored ,as well as histopathological analysis of lungs at the end of the experiment, comparing LTC treated and control groups, nevertheless, there are not references about the actual infection of the animals. Serum concentrations of antibodies to pathogens should be analyzed.

Reviewer #2: This is an interesting work performed by Wheat et al. that evaluated whether activation innate immune induced in the upper respiratory tract by intranasal administration of liposome-TLR complexes (LTC) would generate protection against

bovine respiratory disease complex. This hypothesis is based on a previous work in rodents showing that intranasal administration of LTC protected against lethal infections with bacterial and viral pathogens.

The authors show in experiments performed in vivo that intranasal administration of LTC reduced clinical signs of infection and disease-associated euthanasia rates to BRDC challenge indicating that intranasal administration of LTC was effective in

generating protection against viral and bacterial respiratory tract infections in cattle.

6. PLOS authors have the option to publish the peer review history of their article (what does this mean?). If published, this will include your full peer review and any attached files.

Reviewer #1: No

Reviewer #2: No

---

## [Author Response · Author response to Decision Letter 0]

2 Jun 2020

To PLoS ONE editors and reviewers: 

 Again, we would like to thank the editors and academic reviewers for their timely review of this manuscript. Accordingly, we have provided a revision per both editorial and scientific review herein. It is our hope that this revision will better suit the editorial requirement for PLoS ONE as well as significantly improve the scientific merit of the study described. Specific changes to the manuscript is as follows:

1) The authors should provide some tests about the disease/health status of the animals before use in the assays:

Editors requested authors to provide some tests about the disease/health status of the animals before use in assay. In the revised manuscript, we have expanded two sections of the Materials and Method section that now include how cattle were assessed for both overall health and pre-treatment pathogen burden. See lines 246-250 in the marked-up version. Additionally, see lines 366-373. We have also revised the manuscript to include the results of culturing pathogenic respiratory bacteria from cattle before, during and at the conclusion of the study (day 24): lines 624-630. 

2) Please, answer to all the questions raised by both the reviewers.

See below… 

Revisions towards the Journal requirements: 

Editorial issues are cited in green font and reviewers’ comments are cited as red font.

Editorial Comments #1: Please ensure that your manuscript meets PLOS ONE's style requirements, including those for file naming. The PLOS ONE style templates can be found at

We have checked and revised, where needed, any style requirements per instruction of the editors.

Editorial comments #2: We note that you have included the phrase “data not shown” in your manuscript. Unfortunately, this does not meet our data sharing requirements. PLOS does not permit references to inaccessible data. We require that authors provide all relevant data within the paper, Supporting Information files, or in an acceptable, public repository. Please add a citation to support this phrase or upload the data that corresponds with these findings to a stable repository (such as Figshare or Dryad) and provide and URLs, DOIs, or accession numbers that may be used to access these data. Or, if the data are not a core part of the research being presented in your study, we ask that you remove the phrase that refers to these data.

We feel as though there was no need for any external file sharing since all of the data are presented in the manuscript as written. Please note that we have either removed or provided data each time the phrase “data not shown” was used in the first submitted version of the manuscript: 

Important: All line numbers refer to the “marked up” copy entitled: “Revised Manuscript”

Line 229: “not shown” was deleted since it is not a core part of the research. 

Line 434: “data not shown” was deleted and a revised Figure 2 (Fig 2A) showing the data was included. 

Line 473: “data not shown” was removed because the revised Figure now shows the results of the pretreatment of the macrophages with the nitric oxide inhibitor aminoguanidine (AG)

Line 551: “data not shown” removed because this result was not a core part of the overall project. 

Lines 614-618: “data not shown” phrases were deleted since the data change was not significant or pertinent to the core research described. The sentences describing the marginal changes in cytokine secretion were edited to provide better clarity. 

Line 710 “data not shown” was deleted since the data are now shown in revised Fig. 4. 

Line 757 “data not shown” deletes since data are not a core part of the research. 

Line 620, the authors changed “TNFa” to “IL-6” to correct the typographical error. 

Editorial comments # 3. Thank you for stating the following in the Financial Disclosure section:

"SD was supported by a grant from the State of Colorado Office of Economic

Development and International Translation (OEDIT) and by a grant from Colorado State University Research Council (CRC)#COLV 2018-06. OEDIT is intended to

support the state of Colorado bioscience industry and funds were used herein to fund

materials and salary support for studies, including sample collection, research

supplies, data analysis and publication costs. Animal BRDC challenge studies were

supported by Elanco, Inc.

We note that one or more of the authors are employed by a commercial company: "Hunter Cattle Company".

We note that you received funding from a commercial source: "Elanco, Inc"

a) Please provide an amended Funding Statement declaring these commercial affiliations, as well as a statement regarding the Role of Funders in your study. If the funding organization did not play a role in the study design, data collection and analysis, decision to publish, or preparation of the manuscript and only provided financial support in the form of authors' salaries and/or research materials, please review your statements relating to the author contributions, and ensure you have specifically and accurately indicated the role(s) that these authors had in your study. You can update author roles in the Author Contributions section of the online submission form.

Within your Competing Interests Statement, please confirm that this commercial affiliation does not alter your adherence to all PLOS ONE policies on sharing data and materials by including the following statement: "This does not alter our adherence to PLOS ONE policies on sharing data and materials.” (as detailed online in our guide for authors http://journals.plos.org/plosone/s/competing-interests. If this adherence statement is not accurate and there are restrictions on sharing of data and/or materials, please state these. Please note that we cannot proceed with consideration of your article until this information has been declared.

We have provided both an amended Funding Statement and an updated Competing Interest Statement in accordance with editors’ request. 

Reviewers' comments:

[sic] “Reviewer #1: In this study a series of in vitro and vivo assays were performed to demonstrate the potential of lipossomes complexed to TLR-3 and TLR-9agonists (LTC) to activate upper airway innate imune responses, protecting cattle of infections associated with Bovine Respiratory Disease Complex (BRDC).

The assays were performed in PBMC, PBMC derived macrophages and in naso pharingeal cells from healthy animals, where they demonstrated a dose response activation of the different cells by LTC incubation in culture. Several parameters were evaluated such as IFN-gamma, IL-8, IL-6, MCP1 secretion and/or gene expression of these mediators and increase of class II MHC expression by activated monocytes or macrophages

In PBMC derived macrophages from healthy donors they studied the action of LTC in phagocytosis of beads and of bacteria, as well as the Nitric oxid dependent bactericidal activity following LTC incubation. All these reactions were improved by LTC in a dose dependente manner.

All experiments were well designed and the results converge to the demonstration of the effectiveness of LTC as an alternative preventive or therapeutic agent to BRDC.

Our only concern is about the last in vivo experiment. There it is lacking the confirmation of the infectious status of the experimental groups that were in contact with the seeders carrying the pathogens. Clinical signs were scored ,as well as histopathological analysis of lungs at the end of the experiment, comparing LTC treated and control groups, nevertheless, there are not references about the actual infection of the animals. Serum concentrations of antibodies to pathogens should be analyzed.”

Reviewer 1 requested that in addition to clinical scores that the authors should include data showing the infectious state of the experimental groups that were in contact with seeders. Accordingly, we have introduced a new supplemental figure S3 showing the infectious burden of respiratory pathogenic bacteria (Mannheimia spp, P. multocida or T. pyogenes in the cattle exposed to seeder pre-treatment(pre-exposure), mid-treatment and at the end of the study. Lines 246-250 and 366-373 have now been added to the Materials and Methods section describing the health status and pathogen burden of the animals that were selected for the study. Lines 624-630 are now added to the Results section referenced by a new Supplemental Figure S3 showing the percent of animals with either: no significant, moderate or high bacterial burden from culturing of nasal swabs at the beginning and middle of the project and of lung tissue from necropsies at the end of the study (day 24). 

[sic] Reviewer #2: This is an interesting work performed by Wheat et al. that evaluated whether activation innate immune induced in the upper respiratory tract by intranasal administration of liposome-TLR complexes (LTC) would generate protection against

bovine respiratory disease complex. This hypothesis is based on a previous work in rodents showing that intranasal administration of LTC protected against lethal infections with bacterial and viral pathogens.

The authors show in experiments performed in vivo that intranasal administration of LTC reduced clinical signs of infection and disease-associated euthanasia rates to BRDC challenge indicating that intranasal administration of LTC was effective in

generating protection against viral and bacterial respiratory tract infections in cattle.”

Reviewer 2 made no request for revision, hence, no response from the authors. 

Sincerely yours, 

Dr. William H. Wheat

Department of Clinical Sciences

Colorado State University

Fort Collins, CO 80523-1619

U.S.A.

---

## [Decision Letter · Decision Letter 1]

16 Jun 2020

Non-Specific Protection from Respiratory Tract Infections in Cattle Generated by Intranasal Administration of an Innate Immune Stimulant

PONE-D-20-09236R1

Dear Dr. Wheat,

We’re pleased to inform you that your manuscript has been judged scientifically suitable for publication and will be formally accepted for publication once it meets all outstanding technical requirements.

Kind regards,

Paulo Lee Ho, Ph.D.

Academic Editor

PLOS ONE

Additional Editor Comments (optional):

Reviewers' comments:

Reviewer's Responses to Questions

**Comments to the Author**

1. If the authors have adequately addressed your comments raised in a previous round of review and you feel that this manuscript is now acceptable for publication, you may indicate that here to bypass the “Comments to the Author” section, enter your conflict of interest statement in the “Confidential to Editor” section, and submit your "Accept" recommendation.

Reviewer #1: All comments have been addressed

2. Is the manuscript technically sound, and do the data support the conclusions?

Reviewer #1: Yes

3. Has the statistical analysis been performed appropriately and rigorously? 

Reviewer #1: Yes

4. Have the authors made all data underlying the findings in their manuscript fully available?

Reviewer #1: Yes

5. Is the manuscript presented in an intelligible fashion and written in standard English?

Reviewer #1: Yes

6. Review Comments to the Author

Reviewer #1: (No Response)

7. PLOS authors have the option to publish the peer review history of their article (what does this mean?). If published, this will include your full peer review and any attached files.

Reviewer #1: Yes: Olga M Ibanez

---

## [Editor Report · Acceptance letter]

17 Jun 2020

PONE-D-20-09236R1 

Non-specific protection from respiratory tract Infections in cattle generated by intranasal administration of an innate immune stimulant 

Dear Dr. Wheat:

I'm pleased to inform you that your manuscript has been deemed suitable for publication in PLOS ONE. Congratulations! Your manuscript is now with our production department. 

Kind regards, 

on behalf of

Dr. Paulo Lee Ho 

Academic Editor

PLOS ONE